# Gradual compaction of the nascent peptide during cotranslational folding on the ribosome

**Marija Liutkute[1], Manisankar Maiti[1], Ekaterina Samatova[1], Jörg Enderlein[2], Marina V Rodnina[1]***

[1]Department of Physical Biochemistry, Max Planck Institute for Biophysical Chemistry, Göttingen, Germany; [2]III. Institute of Physics - Biophysics, Georg August University, Göttingen, Germany

**Abstract** Nascent polypeptides begin to fold in the constrained space of the ribosomal peptide exit tunnel. Here we use force-profile analysis (FPA) and photo-induced energy-transfer fluorescence correlation spectroscopy (PET-FCS) to show how a small $\alpha$-helical domain, the N-terminal domain of HemK, folds cotranslationally. Compaction starts vectorially as soon as the first $\alpha$-helical segments are synthesized. As nascent chain grows, emerging helical segments dock onto each other and continue to rearrange at the vicinity of the ribosome. Inside or in the proximity of the ribosome, the nascent peptide undergoes structural fluctuations on the µs time scale. The fluctuations slow down as the domain moves away from the ribosome. Mutations that destabilize the packing of the domain's hydrophobic core have little effect on folding within the exit tunnel, but abolish the final domain stabilization. The results show the power of FPA and PET-FCS in solving the trajectory of cotranslational protein folding and in characterizing the dynamic properties of folding intermediates.

**\*For correspondence:**
rodnina@mpibpc.mpg.de

**Competing interests:** The authors declare that no competing interests exist.

## Introduction

The ribosome synthesizes proteins according to the sequence of the messenger RNA (mRNA) by progressively adding amino acids to the C-terminus of the nascent peptide. Many proteins begin to fold during ongoing translation (*Kramer et al., 2019*; *Rodnina, 2016*; *Waudby et al., 2019*). Translation dictates how much of the nascent chain is available for folding at a given moment. The growing peptide moves from the peptidyl transferase center (PTC) of the ribosome into the peptide exit tunnel, which is ~100 Å long and ~20 Å wide at the vestibule where the nascent peptide emerges into the cytoplasm. The narrowest part of the tunnel – the constriction site – is ~30 Å away from the PTC and about 10 Å wide (*Nissen et al., 2000*). The dimensions of the exit tunnel generally prevent formation of complex tertiary interactions. However, studies using Förster Resonance Energy Transfer (FRET) (*Woolhead et al., 2004*; *Holtkamp et al., 2015*; *Mercier and Rodnina, 2018*; *Khushoo et al., 2011*), cysteine accessibility assays (*Lu and Deutsch, 2005*; *Kosolapov and Deutsch, 2009*; *Tu et al., 2014*), and cryo-electron microscopy (cryo-EM) (*Bhushan et al., 2010*; *Javed et al., 2017*; *Nilsson et al., 2015*; *Su et al., 2017*; *Nilsson et al., 2017*) of ribosome–nascent-chain complexes (RNCs) have shown that some secondary and tertiary structures can form inside specific folding zones of the exit tunnel. The upper part of the tunnel from the PTC to the constriction can accommodate $\alpha$-helices, whereas the lower part of the tunnel, past the constriction, allows secondary and tertiary hairpin structures to form. The ribosome can affect folding in different ways. It can alter the folding pathway of a protein from concerted to sequential, driven by the vectorial appearance of structural elements in the exit tunnel (*Holtkamp et al., 2015*; *Mercier and Rodnina, 2018*); induce helical and random-coil behavior of nascent peptide, alter the folding rates, and

facilitate the correct folding of nascent chains (*Kaiser et al., 2011*; *Liu et al., 2017*; *Alexander et al., 2019*). Upon emerging from the tunnel, the nascent chain can form a near-native compact structure while still bound to the ribosome (*Khushoo et al., 2011*; *Nilsson et al., 2017*; *Waudby et al., 2018*; *Cabrita et al., 2016*; *Kim et al., 2015*; *Samelson et al., 2018*; *Farías-Rico et al., 2018*). The ribosome can significantly destabilize folded globular proteins in its vicinity (*Kaiser et al., 2011*; *Liu et al., 2017*; *Alexander et al., 2019*); the stability can be restored by extending the linker and allowing the protein to move further away from the ribosome surface (*Cabrita et al., 2016*; *Samelson et al., 2016*).

As the peptide folds within the exit tunnel, it exerts mechanical tension (*Nilsson et al., 2015*; *Ismail et al., 2012*), which is transmitted over the length of the nascent chain to the PTC (*Fritch et al., 2018*). In the absence of any additional mechanical force, a 17 amino acids (aa)-long translational arrest peptide, SecM, is sufficient to stall translation (*Ito and Chiba, 2013*). A high-tension mechanical pulling force (7–16 pN) originating from nascent protein folding (*Goldman et al., 2015*; *Kemp et al., 2020*), can alleviate the SecM stalling and restart translation (*Nilsson et al., 2015*; *Ismail et al., 2012*; *Goldman et al., 2015*). This phenomenon is utilized in arrest peptide-mediated force-profile assays (FPA), which monitor force generation events at various stages of cotranslational folding and generate a tension profile that reflects the folding trajectories of proteins (*Nilsson et al., 2015*; *Farías-Rico et al., 2018*; *Ismail et al., 2012*; *Goldman et al., 2015*; *Tian et al., 2018*; *Kemp et al., 2019*). FPA-derived folding trajectories are consistent with FRET, PET, NMR, and cryo-EM analyses of cotranslational protein folding (*Holtkamp et al., 2015*; *Nilsson et al., 2015*; *Cabrita et al., 2016*; *Kemp et al., 2019*). FPA can identify folding events at single-codon resolution, but this potential has not been utilized so far in identifying early folding intermediates inside the peptide exit tunnel.

Among the different approaches to study protein folding, fluorescence correlation spectroscopy (FCS) can provide unique insights into protein dynamics down to nanosecond time resolution (*Neuweiler et al., 2003*). To fully harness the FCS potential and study the local conformational fluctuations of peptide chains, FCS can be combined with fluorescence quenching by photoinduced electron transfer (PET-FCS) (*Neuweiler et al., 2003*). This approach was successfully applied to study protein folding and dynamics in solution (*Neuweiler et al., 2009*; *Neuweiler et al., 2010*; *Schulze et al., 2016*). PET requires short-range interactions (van der Waals contacts,<1 nm) between quencher and dye, which allows to specifically monitor local structural fluctuations of protein chains. Using PET-FCS to monitor cotranslational folding intermediates on the ribosome could provide insight into their structural dynamics and detect rapid local fluctuations between different conformations of the nascent chain, but so far such experiments have not been carried out.

In this work, we apply FPA and PET-FCS to investigate the folding trajectory and conformational fluctuations of the α-helical N-terminal domain (NTD) of an *E. coli* (N5)-glutamine methyltransferase protein HemK on the ribosome. We have chosen the NTD as a model protein because in solution it folds independently of the C-terminal domain and its folding on the ribosome differs dramatically from the two-state concerted folding of the free protein in solution (*Holtkamp et al., 2015*; *Mercier and Rodnina, 2018*; *Kemp et al., 2019*). The cotranslational folding pathway is not known in detail, but FRET and PET studies suggest that cotranslational folding of HemK NTD proceeds sequentially through several compact intermediates (*Holtkamp et al., 2015*; *Mercier and Rodnina, 2018*), consistent with initial FPA results (*Kemp et al., 2019*). Furthermore, previous time-resolved experiments showed that the observed folding dynamics is rapid compared to the pace of translation, and that the nascent chains arrested at different stages of translation (stalled RNCs) faithfully reflect the dynamics that occurs during synthesis (*Holtkamp et al., 2015*). Here, we identify the timing of folding events for the wild-type (wt) NTD and its destabilized 4xA variant, uncover the rates of conformational fluctuations of cotranslational folding intermediates, and define the contribution of the ribosome in maintaining the stability of these compact structures.

## Results

### Sequential folding of HemK NTD on the ribosome

Native folded HemK NTD (72 amino acids (aa)) consists of five α-helices (H1 to H5) arranged into a globular domain, identical to the NTD conformation in the full HemK protein (*Yang et al., 2004*;

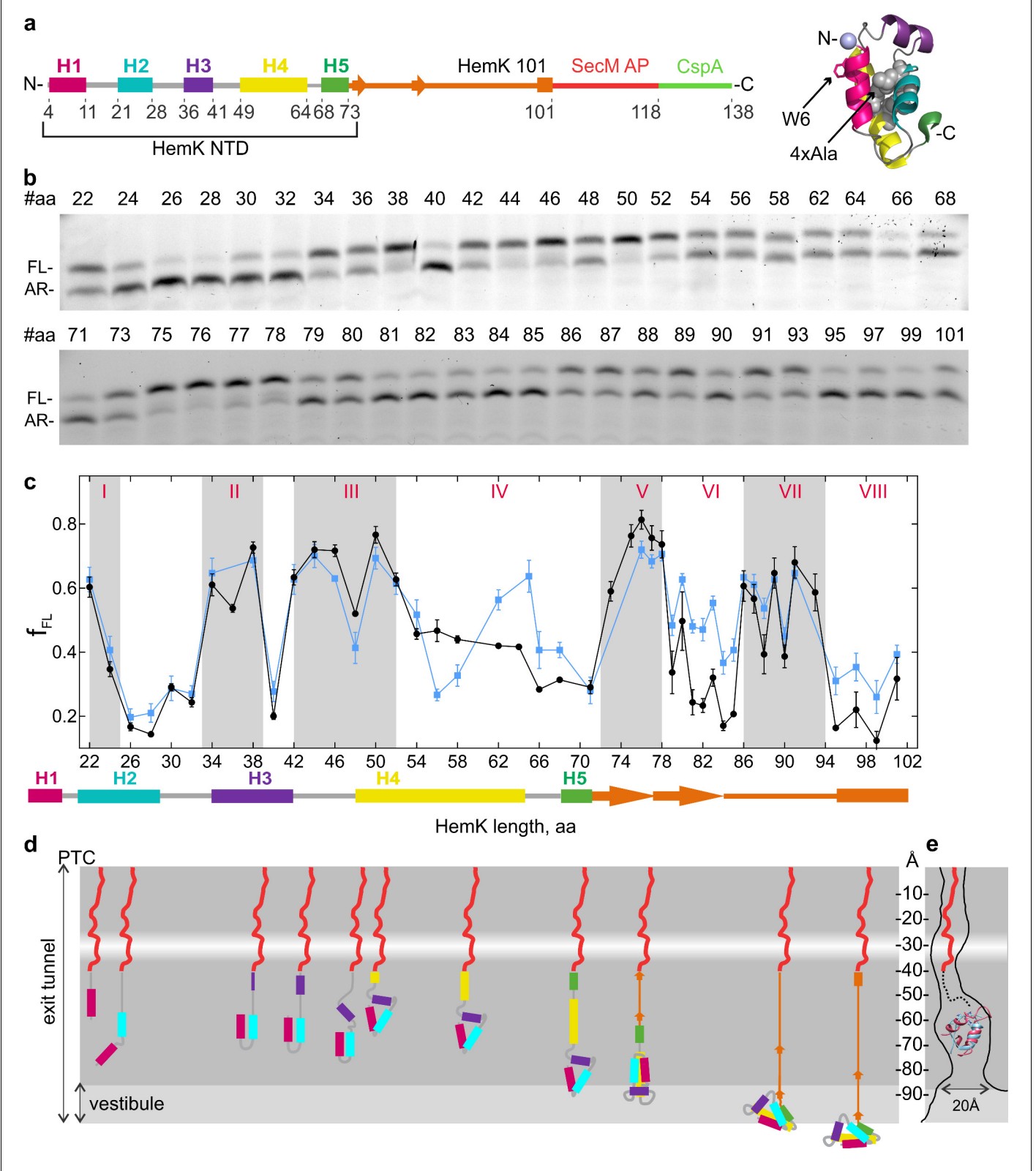

**Figure 1.** Co-translational folding of HemK NTD revealed by high-resolution FPA. (a) Schematic representation of the FPA sensor. HemK NTD helices (H) 1–5, the C-terminal linker (orange), SecM arrest peptide (red), and CspA (green) (left) and crystal structure of HemK NTD (PDB ID: 1T43) (right). The mutations introduced in the 4xA variant (L27A, L28A, L55A, L58A) are shown in gray and the N-terminal fluorophore position is shown in lilac. (b) SDS-PAGE of in vitro translation products for the FPA construct of wt HemK. The length of the nascent chain from the N-terminus to SecM AP is indicated

*Figure 1 continued on next page*

*Figure 1 continued*

(#aa). FL, full-length product; AR, arrested peptide. For controls with shorter nascent chains, see *Figure 1—figure supplement 1d*. (c) Force profile of HemK NTD folding. $f_{FL}$ is the fraction of the full-length product formed during in vitro translation. Black, HemK wt; blue, HemK 4xA mutant; error bars indicate standard error of mean calculated from three independent biological replicates ($N = 3$).The schematic underneath shows the potential secondary structure of HemK at the indicated aa chain length. (d) Schematic diagram of HemK NTD compaction events during translation; color code as in (a). The constriction site is indicated by a white band. (e) Schematic overlay of the HemK H1-H3 crystal structure (pink) (PDB ID: 1T43) and ADR1 Zn finger domain crystal structure (blue) inside the peptide exit tunnel, 60–80 Å from the PTC (*Nilsson et al., 2015*) (PDB ID: 2ADR; EM map: EMD-3079). The online version of this article includes the following source data and figure supplement(s) for figure 1:

**Source data 1.** Numerical values of HemK wt and 4xA variant FPA f_FL.
**Figure supplement 1.** Representative SDS PAGE of FPA for HemK 4xA variant.
**Figure supplement 1—source data 1.** Numerical values of HemK W6F and Pro variant FPA f_FL.

*Figure 1a*). In solution this domain folds at a rate of 5 ms$^{-1}$ while the translation rate for HemK NTD is 3.6 amino acids s$^{-1}$ (*Holtkamp et al., 2015*). To map the cotranslational folding pathway, we performed high-resolution FPA covering aa 22 to 101 of HemK. We generated a construct that encodes aa 1 to 101 of HemK, which includes the NTD (aa 1–73) and part of the interdomain linker connecting the NTD to the C-terminal domain, followed by 17 codons for the SecM arrest peptide, and an additional sequence encoding 20 aa of protein CspA; the latter served as a reporter for high-tension events in the nascent chain. With the HemK101 construct, the resulting nascent chain (including the SecM peptide; 118 aa in total) is long enough for the entire NTD to emerge from the ribosome. At low force, the ribosome is stalled by the SecM arrest peptide, generating an arrested translation product (AR) (*Nilsson et al., 2015*). If folding of HemK exerts force, translation arrest is alleviated, resulting in a longer peptide, which we denote as full-length (FL) (*Figure 1b*). We then constructed a series of mRNAs where the HemK sequence is trimmed in steps of one or two codons from the 3′ end of the HemK mRNA. We translated these mRNAs in a fully reconstituted in vitro translation system from *E. coli*. In this system, the ribosomes are synchronized at the initiation step and start translation simultaneously after mixing with elongation factors and aa-tRNAs. The experimental setup allows for a single round of translation on a given mRNA, thereby avoiding the potential desynchronization due to re-initiation. The translation products, AR and FL, are separated by SDS-PAGE. By analyzing the fraction of FL product formed, we identify high-tension folding events (*Figure 1c*).

The detailed force profile of the HemK NTD reveals several distinct force-generating folding events starting early inside the exit tunnel until the entire NTD emerges from the ribosome (*Figure 1c,d*). The early high-tension regions are observed at nascent-chain lengths of aa 22–24 (I), 33–39 (II) and 42–52 (III) with a transient force relief at aa 48. At these chain lengths, the nascent peptide is likely to reside in the exit tunnel, as it is protected from protease digestion (*Holtkamp et al., 2015*). To test whether high-tension regions reflect folding events, we generated a variant of HemK with proline mutations in N-terminal helices H1-H3, which should prevent native helix formation and disturb native cotranslational tertiary packing of the nascent chain. Introducing prolines alters the FPA profile, but to a somewhat different extent in regions I, II and III (*Figure 1— figure supplement 1f,g*). At 22–24 aa (region I), the wt nascent peptide entails H1 and the subsequent loop region that moved past the tunnel constriction; hence, the observed high tension can reflect folding of H1 on its own or together with the first helical turn of H2. Introducing prolines into the 22–32 region leads to a flattening of the FPA profile (*Figure 1—figure supplement 1g*), consistent with the notion that the characteristic FPA profile provides a signature for sequence-specific protein compaction within confines of the tunnel. Furthermore, if the polypeptide is truncated within the loop between H1 and H2, tension is abolished (*Figure 1—figure supplement 1d*), suggesting that the entire first loop sequence together with H1 are necessary to create the high-tension event. This very early cotranslational intermediate of HemK folding deep inside the exit tunnel was not observed so far. The force level decreases as more of H2 emerges and H1 moves further down the exit tunnel, and then increases again before the complete helix H3 is synthesized (high-tension region II). Once H1 and H2 move toward the more open space of the vestibule, they may begin to form tertiary interactions, thereby generating tension. Proline substitutions in the 34–38 region

decrease tension considerably (*Figure 1—figure supplement 1g*), supporting the notion that region II likely represents a folding event.

Region III at aa 42–52 (*Figure 1c,d*) broadly coincides with the compacted intermediate identified by FRET and FPA studies (*Holtkamp et al., 2015*; *Mercier and Rodnina, 2018*; *Kemp et al., 2019*). The tension increases as the entire H3 emerges below the constriction and most likely corresponds to the formation of H3 and its docking onto the preceding two-helix structure. Proline mutations abolish formation of the high-tension peak (*Figure 1—figure supplement 1g*). The transient tension relief at 48 amino acids may separate the helix formation and docking events. The H1-H3 intermediate structure is expected to reside approximately 50–60 Å away from the PTC in the region within the exit tunnel that is known to accommodate a folded small zinc-finger domain protein (ADR1) (*Nilsson et al., 2015*). To validate the feasibility of the HemK intermediate forming in this region of the tunnel we have superimposed the HemK H1-H3 onto the structure of the ADR1 domain in the tunnel (*Nilsson et al., 2015*). The two domains are of similar size (*Figure 1e*), and even if the HemK H1-H3 occupies a slightly larger volume than ADR1, it is very likely that the folding intermediate of H1-H3 can be accommodated in this region of the tunnel. The analysis of the folding regions I-III provides the first example of multiple cotranslational folding intermediates being resolved in the exit tunnel.

As HemK becomes longer than 53 aa, the tension decreases to intermediate values, but still remains above the baseline tension level (~0.2) (*Figure 1c*, region IV). Protease protection experiments suggest that when the nascent chain reaches the total length of 84 aa, position 34 (fluorescently labeled in those experiments) is no longer protected by the ribosome (*Holtkamp et al., 2015*). In the FPA experiments, this total chain length corresponds to 67 aa of HemK, of which at least 34 aa must be exposed to the solvent. At the HemK NTD length of 72 aa (99 aa total), part or all of H4 may emerge from the vestibule and be ready to initiate the formation of further hydrophobic interactions in the HemK NTD (*Figure 1d*). The next high-tension intermediate at aa 72–78 (region V) forms when H5 and part of the following interdomain linker move past the constriction and H4 is further displaced toward the vestibule. A stable folding of H4 within the exit tunnel may form hydrophobic interactions with the H1-H3 structure, which generates the high-tension peak because the structure is closer to a narrow part of the vestibule. As the nascent chain grows, the H1-H4 intermediate again moves away from the exit port, which results in further structural rearrangements reflected in the decrease of tension interrupted by short force spikes at aa 80 and 83 (region VI).

The final high-tension compaction (step VII) occurs when the length of the HemK NTD exceeds 86 aa, placing the entire NTD, H1 through H5, outside of the confines of the exit tunnel (*Figure 1c,d*). When NTD length is >95 aa (112 aa total peptide length), we observe only basal tension levels, indicating that the final cotranslational folding to a near-native conformation of the NTD occurred at step VII, consistent with the final steps of NTD cotranslational folding suggested previously (*Holtkamp et al., 2015*; *Mercier and Rodnina, 2018*).

In addition to the wt HemK NTD, we examined cotranslational folding of the destabilized HemK NTD variant (4xA), where four conserved Leu residues of the hydrophobic core (Leu 27 and 28 in H2, and 55 and 58 in H4) were mutated to Ala (*Figure 1a*; *Holtkamp et al., 2015*). The force profile inside the exit tunnel is identical for the wt and the 4xA variant (folding steps I-III), but as the peptide begins to emerge at the vestibule, the force profiles start to deviate (*Figure 1c*, *Figure 1—figure supplement 1a,e*). As more of H4 becomes accessible for docking to the H1-H3 structure, Leu-specific interactions within the hydrophobic core start to matter. For the 4xA variant, we observe a drop in tension at a total length of 73 aa (56 aa of HemK NTD), and the appearance of a high-force intermediate at a chain length where the tension in the wt HemK NTD gradually decreases (aa 62–64) (region IV). The change in folding at region IV could result from a delay (of four aa) in folding for the 4xA variant compared to the wt. This could be because in the absence of the key Leu residues in the hydrophobic core a longer peptide is required to stabilize the packing, whereas the wt variant folds continuously as more of the nascent chain becomes available. Alternatively, it is possible that this peak indicates an intermediate state unique to 4xA.

The two following cotranslational folding steps V and VII are similar in duration and amplitude for 4xA and wt, suggesting that these rearrangements are independent of the hydrophobic core packing. In contrast, in regions VI and VIII, the 4xA variant generates consistently higher tension than the wt NTD. This agrees with the notion that the 4xA variant adopts an expanded conformation; the

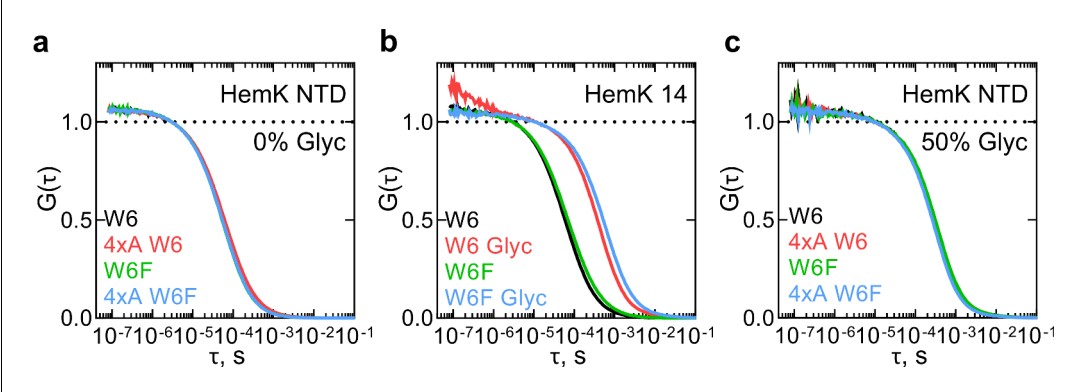

**Figure 2.** Monitoring dynamics of HemK constructs in solution by PET-FCS. (**a**) ACF of the HemK NTD free in solution. Black, wt variant with Trp at position 6 (W6); red, destabilized 4xA HemK W6 variant; green, HemK wt with no Trp in the nascent chain (W6F); blue, 4xA HemK W6F variant; each curve is derived from two separate release experiments and each experiment consists of at least four technical replicates ($N \geq 8$). (**b**) Autocorrelation curves of HemK14 peptide in solution. Black, W6 variant at low solvent viscosity; red, W6 at high viscosity in the presence of 50% glycerol (Glyc); green, W6F variant at low viscosity; blue, W6F variant at high viscosity. Shown are representative curves of at least two experimental repeats, each consisting of at least four technical replicates ($N \geq 8$). (**c**) Autocorrelation curves of HemK NTD released from the ribosome at high viscosity conditions (50% glycerol). Black, wt W6; red, 4xA W6; green, wt W6F; blue, 4xA W6F. Each curve is derived from two separate release experiments and each experiment consists of at least four technical replicates ($N \geq 8$).

The online version of this article includes the following source data for figure 2:

**Source data 1.** Numerical values of the calculated autocorrelation functions of HemK constructs in solution.

timing of the discrete rearrangements that occur at step IV appears similar for 4xA and wt NTD, but each time when a more bulky 4xA peptide moves toward the ribosome surface, the tension increases. This difference in tension may indicate that the non-native 4xA fold occupies a larger volume has a more dynamic structure, or there are changes in domain stability (*Leininger et al., 2019*).

## Nascent-chain dynamics monitored with PET-FCS

Next, we monitored the dynamics or ribosome-bound nascent chains by PET-FCS. We utilized the native Trp6 residue (W6) in HemK that could quench an N-terminal ATTO655 fluorophore upon coming into van der Waals distance (*Doose et al., 2009*). The dynamic motions of the nascent chain define the frequency of these quenching interactions, and the resulting fluorescence intensity fluctuations can be used to track the internal dynamics of the nascent chain (*Neuweiler et al., 2003*).

First, we studied the dynamics of the HemK NTD in solution. We prepared a 70 aa-long NTD wt or 4xA by in vitro translation using an ATTO655-labeled initiator Met-tRNA$^{fMet}$, purified the complexes, and then chemically released the nascent chains from the ribosome (Materials and methods). As a control for the PET signal, we also prepared NTD variants where Trp6 is replaced with non-quenching Phe (W6F). The isolated W6F HemK NTD variant is 90% folded at 37°C compared to the wt and has the same translation rate and profile (*Holtkamp et al., 2015*); the matching cotranslational folding trajectory was further confirmed with the FPA assay (*Figure 1—figure supplement 1b, c*). FCS measurements of the isolated NTD showed a stable and highly reproducible diffusion time of the peptide ($\tau_{d1} \approx 6 \times 10^{-5}$ s) (*Supplementary file 1* - table 1). However, no PET signal was observed, as the autocorrelation functions (ACF) with W6 and W6F proteins are identical for wt and 4xA constructs (*Figure 2a*). This result indicates that there are no dynamic fluctuations at the N-terminus of the HemK NTD in solution. To test this notion, we performed control measurements with a short HemK peptide of 14 aa, which should be unstructured in solution (*Figure 2b*). After testing different conditions, we were able to observe fast dynamics due to PET with a quenching relaxation time of ~$5 \times 10^{-7}$ s in the presence of glycerol (*Figure 2b*, *Supplementary file 1* - table 1). However, when the experiments with HemK NTD wt or 4xA are repeated at the same conditions, we observe only an increase in the diffusion time ($\tau_{d1}$ = ~$30 \times 10^{-5}$ s), but no PET signal corresponding to structural fluctuations (*Figure 2c*, *Supplementary file 1* - table 1). Thus, in solution, HemK NTD, as well as its destabilized 4xA variant, folds into a domain where W6 is not accessible for interaction with the N-terminal dye.

We next applied PET-FCS to study dynamics of nascent peptides. We chose the peptide lengths at which all of the NTD has been synthesized and generated RNCs at three stages of translation based on the FRET and PET measurements (*Mercier and Rodnina, 2018*; *Figure 3a*). While we indicate the approximate FPA profile peak that these RNC constructs will correspond to, the all-native chain dynamics might differ from the nascent peptides in the FPA assay due to the presence of non-native SecM arrest sequence at the C-terminus (17aa) (*Figure 1a*). To monitor the nascent chain dynamics during the compaction of H1-H3 within the exit tunnel, we used RNC with a 70 aa-long nascent chain (HemK 70), which shows a high FRET signal and maps roughly between FPA regions III and IV (*Figure 1c,d*). An RNC with the 102 aa-long nascent chain (HemK 102) should expose H1-H4 at the ribosome surface and maps onto the end of FPA region VI; earlier PET measurements indicate a folding transition around this chain length (*Mercier and Rodnina, 2018*). To monitor the fully emerged domain, we used an RNC with a 112 aa-long nascent chain (FPA region VIII; *Figure 1c,d*). To understand how destabilization of the hydrophobic core of the NTD affects the nascent-chain dynamics, we also used the respective 4xA variants. One complication of the PET-FCS experiments with ribosome complexes is that the Trp residues in ribosomal proteins and guanines of the rRNA also quench the N-terminal fluorophore (*Doose et al., 2009*). To account for these interactions of the nascent peptide with the ribosome surface, we compare the W6 and the respective W6F control constructs.

For all RNCs tested, PET-FCS experiments yield multiphasic ACFs spanning the timescale from ms to ns. The diffusion time of the RNC is in the ms time range (*Samelson et al., 2016*). Initial exponential fitting reveals at least three fast dynamic components in the µs time range. One of the components most likely reflects the triplet state pumping and relaxation of the ATTO655 dye in the complex with the ribosome (*Figure 3—figure supplement 1*). The relaxation time of the triplet state of ATTO655 attached to the peptide on the ribosome, about 40 µs, is somewhat larger than that measured with a model peptide, which is in the range of 2 µs (*Luitz et al., 2017*). We then fitted the ACF curves (*Figure 3b–d*) using a combination of two exponential decays, the triplet state

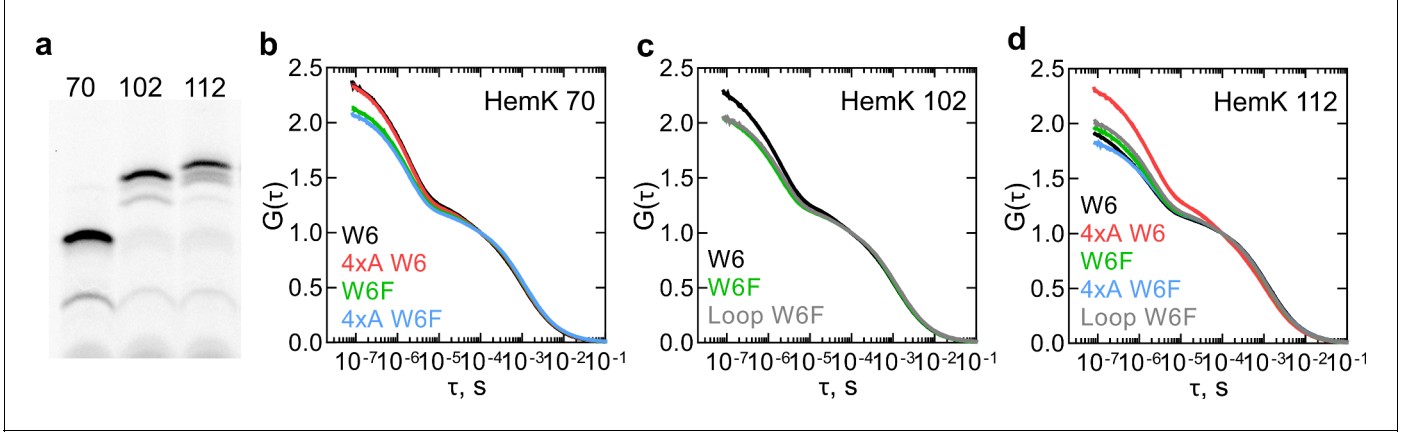

**Figure 3.** Dynamics of HemK on the ribosome monitored by PET-FCS. (**a**) SDS PAGE of nascent chains produced by in vitro translation visualized after RNC purification using the fluorescence of ATTO655. The aa length of HemK wt constructs is indicated. (**b**) Autocorrelation curves for HemK 70-RNC. Black, wt W6; red, 4xA W6; green, wt W6F; blue, 4xA W6F. Each curve is derived from at least two separate RNC preparations and each experiment consists of at least four technical replicates ($N \geq 8$). (**c**) Autocorrelation curves for HemK 102-RNC. Black, wt W6, green, wt W6F; gray, W6F variant with loop extension between helix 3 and helix 4. Each curve is derived from at least two separate RNC preparations and each experiment consists of at least four technical replicates ($N \geq 8$). (**d**) Autocorrelation curves for HemK 112-RNC. Black, wt W6; red, 4xA W6; green, wt W6F; blue, 4xA W6F; gray, W6F variant with a loop extension between helix 3 and helix 4. Each curve is derived from a minimum of two separate RNC preparations and each experiment consists of at least four technical replicates ($N \geq 8$).

The online version of this article includes the following source data and figure supplement(s) for figure 3:

**Source data 1.** Numerical values of the calculated autocorrelation functions of HemK variants on the ribosome.
**Figure supplement 1.** ATTO 655 triplet state in RNC.

correction, and the diffusion term (Materials and methods). The two relaxation times ($\tau = 1/k$) as estimated by fitting, one in the tens of µs and the other in the µs time range, were observed in all tested complexes, regardless of the presence of the W6 residue that causes intra-chain PET (*Supplementary file 1* - table 2). However, the $\tau_1$ and $\tau_2$ values and the respective amplitudes are different for the respective W6 and W6F constructs, indicating that part of the effect is due to intra-chain PET. Taking into account the known timescales of peptide dynamics in solution (*Neuweiler et al., 2010*; *Luitz et al., 2017*; *Lum et al., 2012*), the shorter relaxation time $\tau_1$ most likely reflects the quenching interactions of the fluorophore with the internal Trp or the quenchers at the ribosome surface. The slower relaxation time $\tau_2$ is usually attributed to the dynamics of conformational intermediates. To challenge this assignment, we designed two additional RNCs where we changed the dynamics of peptide chain rearrangements. Because long and unstructured loops are known to enhance conformational fluctuations through increased entropy of the folded state (*Dagan et al., 2013*), we extended a loop between H3 and H4 by five additional Gly residues and generated two RNCs with different lengths of this construct, called 102 loop and 112 loop, respectively. Both $\tau_1$ and $\tau_2$ values are affected by the loop mutations (*Supplementary file 1* - table 2). Because $\tau_1$ and $\tau_2$ are apparent values that have no biological meaning as such, in the following we develop a kinetic modeling approach to determine the elemental rates of nascent-chain dynamics.

## Kinetic model for the dynamics of nascent chains

To fit the ACF data, we examine a series of kinetic models starting with a simple two-step model and going through a number of alternative models of increasing complexity that would describe the whole dataset with the minimum number of parameters. We find that the minimal model that can fit our ACF dataset must include five-equilibria (model 5e-H and model 5e-O) and models with fewer equilibria are insufficient (*Figure 4—figure supplement 1*) (Materials and methods). To account for the exponential term on the µs time scale, we assume that nascent chains can undergo a conformational change from state C to state D (C↔D) (*Figure 4*). In each of these states the N-terminal dye can interact with the internal Trp6. The resulting quenched states are denoted as Wx, leading to the equilibria C↔Wc and D↔Wd. Given that these are fast PET interactions we model the quenched states separately without an equilibrium between them to limit the number of free fitted parameters. By analogy, a quencher on the ribosome surface can quench the fluorescence of states C or D, yielding a non-fluorescent state Rx and the equilibria C↔Rc and D↔Rd (model 5e-H; *Figure 4a*, right panel). An alternative five-equilibria model is possible (model 5e-O) (*Figure 4—figure supplement 2*), where the quenched states are not unique, i.e. the Wx and Rx states are the denoted as the same state W or R (*Figure 4—figure supplement 2*, right panels), this model allows for the possibility of moving from state C to state D and vice versa via one of the shared quenched states W or R.

We perform a global fit of the entire HemK RNC dataset to these models in order to determine the elemental rates of the outlined interactions. For reliable fitting of the µs-range data, we extract the dynamic component from the raw ACF curves by removing from each experimental ACF the diffusional and triplet state components using parameters of the empirical fit (Methods, *Equation 1*; *Supplementary file 1* - table 2). This results in time courses with two exponential decays (*Figure 4*). Because there is no solution to calculate elemental rates from the two apparent rate constants, we additionally measured ACF of RNC 70 W6F and RNC 102 W6F at different free Trp concentrations (*Figure 4—figure supplement 3*). Global fit of the Trp titration data to the five-equilibria models (Materials and methods) yielded ATTO655–Trp dequenching rate of about 2 µs$^{-1}$ (*Supplementary file 1* - tables 3 and 4). These values were then fixed during global fitting of the main dataset (*Figure 4* and *Figure 4—figure supplement 2*), which allowed us to obtain statistically significant values for rate constants describing nascent-chain dynamics of each HemK NTD variant on the ribosome using both five-equilibria models (*Figure 4*; *Figure 4—figure supplements 2* and *4*; *Supplementary file 1* - tables 5, 6 and 7; Materials and methods). Here we will primarily discuss the fitting results of the kinetic model 5e-H, which separates the fast PET-based interactions between dye and quencher, from the slower structural conformational fluctuations of the nascent chain itself, thus more adequately exposing the total extent of conformational fluctuations. It is worth noting that while the obtained absolute rate values are different between models 5e-H and 5e-O, the general trends revealed by the rate analysis are supported by both models (*Figure 4* and *Figure 4—figure supplement 2*; *Supplementary file 1* - tables 5 and 6).

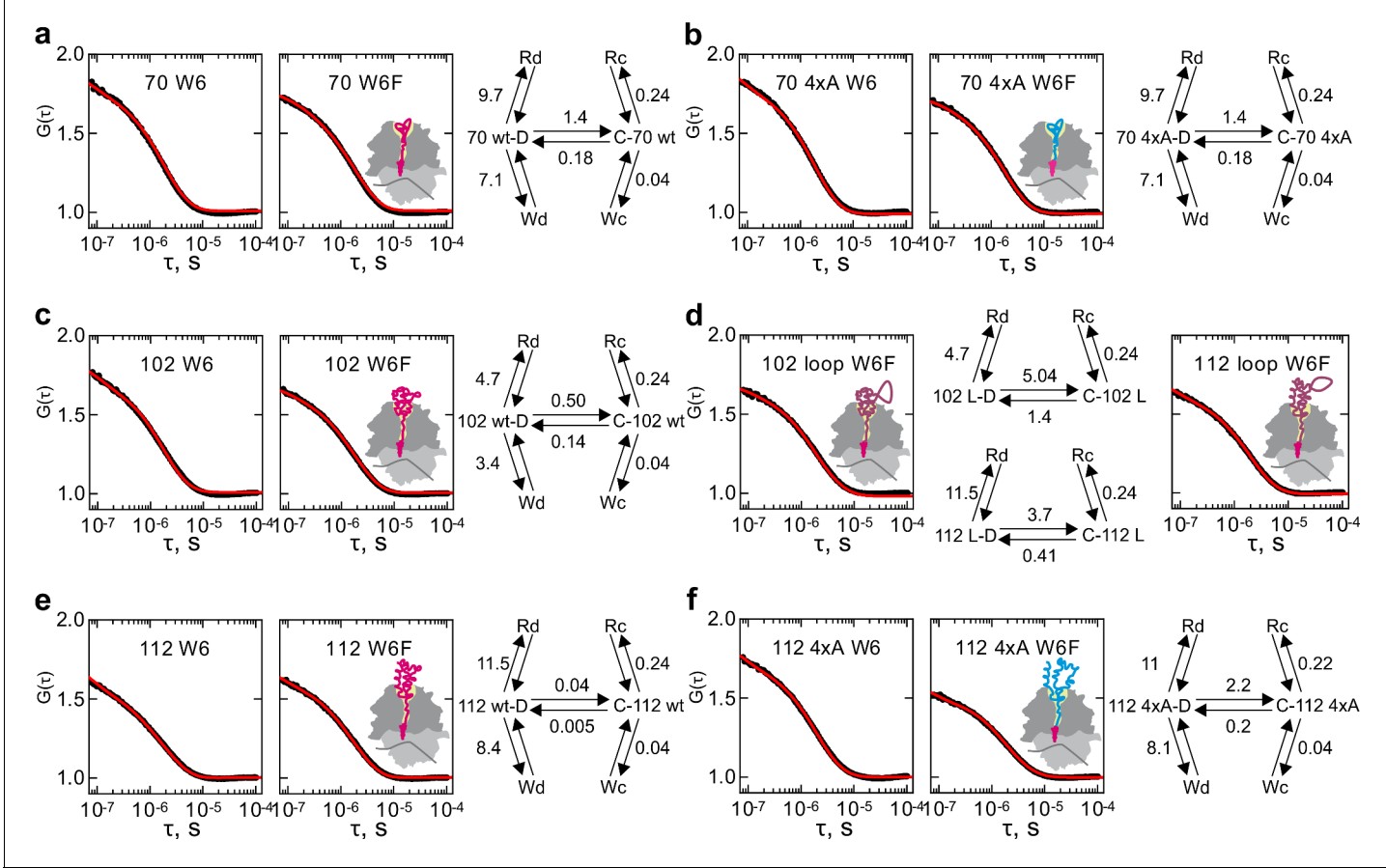

**Figure 4.** Conformational dynamics of RNCs carrying various nascent chains. Shown are the results of global fitting of autocorrelation data to the kinetic model 5e-H (right panel). Black – measured data; red – kinetic model simulation curve. C and D are two peptide conformations; Wx –Trp-quenched states; Rx – ribosome-quenched states. Cartoons indicate the presumed position of the nascent chains on the ribosome, with the wt nascent chains shown in magenta, without and with an extra loop, and the 4xA variant shown in blue. (a) HemK 70 W6 and W6F. (b) HemK 70 4xA W6 and W6F. (c) HemK 102 W6 and W6F. (d) HemK 102 and 112, both W6F, with loop extensions. (e) HemK 112 W6 and W6F. (f) HemK 112 4xA W6 and W6F. The online version of this article includes the following figure supplement(s) for figure 4:

**Figure supplement 1.** Global fitting of HemK RNC dataset B to kinetic models 2e and 3e.
**Figure supplement 2.** Results of global fitting of HemK dataset B to kinetic model 5e-O.
**Figure supplement 3.** Global fitting of autocorrelation curves for HemK W6F RNCs with increasing free Trp concentration.
**Figure supplement 4.** Plots of residuals from the kinetic model 5e fittings of the HemK RNC dataset B.

Comparison of the rate constants for different complexes reveals, first, that the two structural states of the nascent chain, C and D, are very different with respect to their dynamics (*Figure 4*; *Supplementary file 1* - table 5). The rate constants of ATTO655 quenching by the ribosome is 20–50-fold slower in C compared to D in different RNCs. Likewise, quenching by intramolecular Trp6 is 80–200-fold slower in C than in D. We attribute this to lower dynamics of state C and thus denote it as compact, compared to state D, which we call dynamic. Thus, we identify two global conformational ensembles of ribosome-bound HemK NTD.

The second observation is that for a given conformational state C or D, the rates of interaction with Trp or the ribosome are almost insensitive to the extent of folding or the position of the N-terminus on the ribosome, as the $k_{on}$ value for Trp quenching is about 0.04 $\mu s^{-1}$ for state C and 3–8 $\mu s^{-1}$ for state D of all constructs. The low quenching rate constant in the compact states suggests that Trp is confined in the stable hydrophobic core of the respective intermediates. In contrast, the

dynamic state presents a less compact structure wherein the more exposed native Trp interacts more frequently with the N-terminal dye. It is possible that the position of the N-terminus is defined at early stages of folding, which explains why it does not change with the peptide length over 70 aa. More surprisingly, the $k_{on}$ for ribosome quenching is not strongly dependent on the position of the nascent chain with respect to the ribosome, with $k_{on}$ = 0.24 $\mu s^{-1}$ for state C and ranging from 5 $\mu s^{-1}$ to 11 $\mu s^{-1}$ for state D. The dequenching rate of the ATTO655–ribosome complex is 0.3 $\mu s^{-1}$ and identical for 5e-H and 5e-O kinetic models (*Supplementary file 1* - tables 5 and 6).

The largest difference related to the stage of nascent peptide folding and its position on the ribosome pertains to the equilibrium between states C and D. The compact state C is favored in all complexes, but the rate constants of the C↔D transitions differ dramatically, decreasing with increasing protein length and its emergence outside the ribosome (*Figure 4* and *Supplementary file 1* - table 5). The 4xA mutations in the hydrophobic core do not alter the equilibrium between the C and D states (HemK70 and HemK112), but when the domain emerges from the ribosome the mutations cause several orders of magnitude increase of the rate constants between these states (HemK112) (*Figure 4*). Furthermore, introducing a loop of 6 aa dramatically increases the rates of transitions in both HemK102 and HemK112 (*Figure 4*). Thus, as predicted (*Dagan et al., 2013*), the loop extensions increase the helix dynamics of the nascent chain, but do not alter the interaction with Trp or the ribosome surface.

## Discussion

In this work, we combined FPA and PET-FCS to reconstruct the trajectory of cotranslational folding and to evaluate the stability of the folding intermediates. The high-resolution FPA data show that nascent chains undergo sequential force-generating rearrangements that start inside the exit tunnel. The earliest detected force-generating folding intermediate entails as little as a single helix (H1) of the HemK NTD. Upon continued synthesis, emerging helices begin to interact with one another forming tertiary intermediates inside the peptide exit tunnel (*Figure 1d*). The formation of individual α-helices and tertiary interactions between α-helical elements within the exit tunnel are well documented (*Lu and Deutsch, 2005*; *Bhushan et al., 2010*; *Nilsson et al., 2015*; *Farías-Rico et al., 2018*; *Nissley and O'Brien, 2018*). Our results are consistent with the notion that folding of HemK NTD is sequential (*Mercier and Rodnina, 2018*) and show several tension-generating steps corresponding to folding intermediates inside and outside of the ribosome (*Kemp et al., 2019*). In principle, high-tension regions in FPA profiles can arise not only upon protein compaction, but also as a result of interactions between nascent chain and the exit tunnel walls or nascent-chain dynamics within the tunnel. However, the observation that proline substitutions, that are expected to alter helix formation, also reduce the tension suggest that the FPA high-tension regions reflect folding events inside the exit tunnel (*Figure 1d* and *Figure 1—figure supplement 1g*). Changes in folding trajectories upon perturbing protein structure were also demonstrated by FPA for other proteins (*Lu and Deutsch, 2005*; *Bhushan et al., 2010*; *Nilsson et al., 2015*; *Farías-Rico et al., 2018*; *Nissley and O'Brien, 2018*). FPA studies also demonstrate that the growing nascent chain continues to undergo structural adjustments after emerging from the exit tunnel. Some, but not all, of these rearrangements are sensitive to the packing of the protein's hydrophobic core. In contrast, folding of the HemK NTD in solution is concerted, with only two discernible states, native and unfolded (*Holtkamp et al., 2015*). Thus, sequential addition of amino acids during translation affects nascent-protein folding not only inside the exit tunnel, but also at the surface of the ribosome and results in a complex folding pathway with multiple folding intermediates.

Rapid sequential cotranslational folding observed here for HemK NTD can be rationalized using the concept of folding via cooperative folding units, foldons, which was suggested for several proteins based on a combination of NMR, mass spectrometry, and hydrogen-exchange pulse-labeling experiments (*Walters et al., 2013*; *Hu et al., 2013*; *Bai et al., 1995*). The emerging view is that the foldons, comprised of one to two secondary structure elements, form rapidly (e.g. at a rate of 2000 $s^{-1}$ in RNaseH; *Hu et al., 2013*), and once a foldon is formed, the protein undergoes a series of fast folding steps, with native-like foldons added rapidly at each step. The folding trajectory of a particular protein is determined by the nature of foldon units, because each preceding unit guides and stabilizes the subsequent foldons in a thermodynamically downhill energy landscape (*Englander and Mayne, 2014*). The vectorial emergence of nascent peptide into the constrained space of the exit

tunnel may allow nucleation of such folding units and restrict the number of potential interactions/conformations at a given chain length, thereby guiding folding through a relatively narrow energy landscape. This would also explain how folding of the HemK NTD on the ribosome is sequential, guided by the vectorial appearance of foldons during translation. In solution folding is concerted because formation of local folding units defines the rate of the concerted collapse into the native structure.

The PET-FCS experiments show that nascent peptides are dynamic and undergo internal conformational rearrangements on a μs time scale. We identify two subpopulations of folded nascent chains corresponding to the predominant compact and dynamic states, which differ in their ability to interact with the local environment. The local dynamics of the N-terminus as monitored by intramolecular PET between the N-terminal ATTO655 and Trp6 is relatively slow for the compact state (0.04 $\mu s^{-1}$), which resembles the native folded state that is static on the time scale of the FCS experiment, but can unfold on a seconds time scale. The dynamic state has quenching (3–8 $\mu s^{-1}$) and dequenching (about 2 $\mu s^{-1}$) rates similar to those reported for the dynamic motion of proteins (*Neuweiler et al., 2009*; *Neuweiler et al., 2010*; *Luitz et al., 2017*; *Lum et al., 2012*; *Stanley et al., 2014*); the structure of the D state is not known. The quenching interaction with the ribosome is also slow for the compact (0.2 $\mu s^{-1}$) and fast for the dynamic (5–11 $\mu s^{-1}$) state; the dequenching rate is 0.3 $\mu s^{-1}$. The latter rates differ from the dequenching of the ATTO655–Trp complex and most probably reflect the interactions of ATTO655 with guanine residues in rRNA. This is the first time these interactions have been characterized in a context of an RNA-containing macromolecule.

During folding in solution, the rates of fluctuations slow down several-fold as proteins advance from unfolded toward more compact conformations (*Nettels et al., 2007*; *Waldauer et al., 2010*). On the ribosome, the rates of fluctuations between the compact and dynamic states decrease as the nascent chain moves down the exit tunnel and emerges from the ribosome (*Figure 5a*). To compare the nascent chains of different length and stability among each other and demonstrate how the transition state ($\Delta G^{\ddagger}$) free-energy barrier between the compact and dynamic states can change, we used Eyring's transition-state theory (*Fersht et al., 1999*; *Zhou, 2010*). This approach demands some critical assumptions, such as that in solution the initial and the transition states are at a thermal equilibrium, and that once reached the transition-state barrier is crossed in a single step (*Zhou, 2010*; *Hagen, 2010*; *Finkelstein and Ptitsyn, 2016*) and can be considered as an initial simplified description of folding intermediates.

Comparison of the free energy of the transition state ($\Delta G^{\ddagger}$) barrier between the compact and dynamic states at different nascent-chain lengths suggests that as the nascent chain grows,

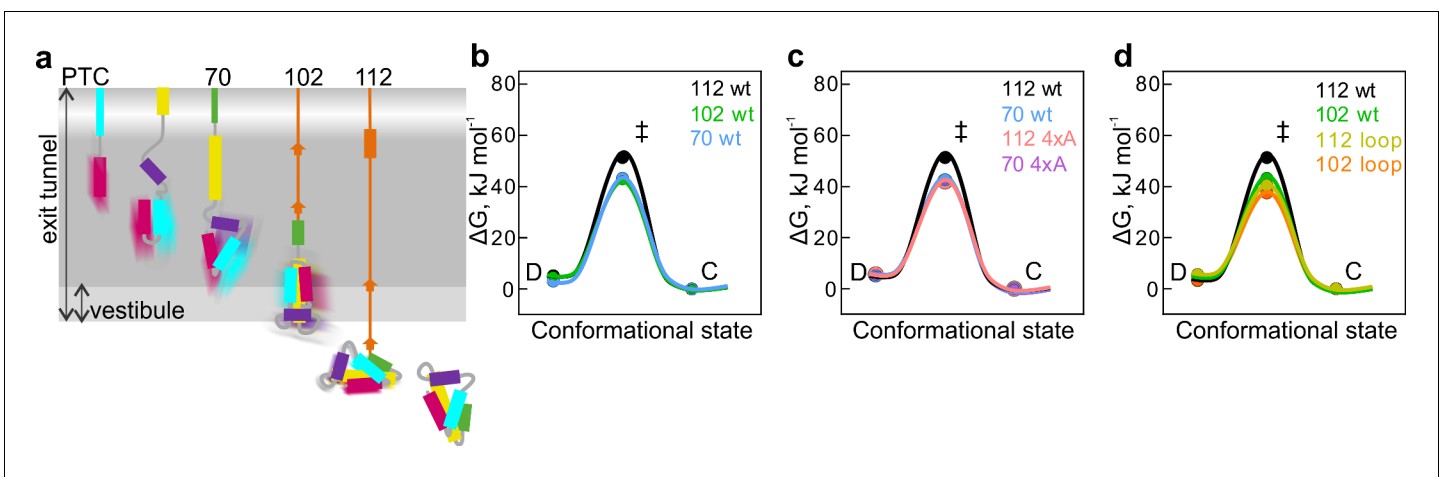

**Figure 5.** Free-energy barriers between different nascent-chain conformations D-dynamic state; C – compact state; ‡ - transition state. Error bars are smaller than data points and represent propagated SEM from elemental rates derived from the kinetic fits to model 5e-H. (**a**) Schematic diagram of cotranslational folding steps of HemK NTD with a visualization of nascent-chain dynamics. (**b**) HemK wt constructs of increasing length, 70 aa (blue), 102 (green), and 112 (black). (**c**) HemK wt and 4xA variants, 70 wt (blue) 70 4xA (lilac), 112 (black), and 112 4xA (red). (**d**) HemK wt and loop variants, 102 wt (green), 112 wt (black), 102 loop (orange), and 112 loop (yellow).

transition-state free energy increases from 38 kJ mol$^{-1}$ for HemK70 to 40 kJ mol$^{-1}$ for HemK102 and to 46 kJ mol$^{-1}$ for HemK112 (*Figure 5b*, *Supplementary file 1* - table 8). The increase in $\Delta G^{\ddagger}$ as the NTD moves away from the ribosome demonstrates the manner in which the proximity of the ribosome may alter the dynamics of the nascent protein domain. Consistent with these results, isolated HemK NTD did not show any PET dynamics. Although the protein can unfold in solution (at a rate of 1 s$^{-1}$ at 37°C), such dynamics is probably too slow for the FCS and the unfolded state is too rare to be detected due to rapid folding (2000 s$^{-1}$) (*Holtkamp et al., 2015*). These data provide further support to the notion that the stability of protein domains increases with the distance to the ribosome (*Kaiser et al., 2011*; *Liu et al., 2017*; *Alexander et al., 2019*; *Cabrita et al., 2016*; *Samelson et al., 2016*) and provide estimations for the rates of rapid conformational fluctuations of nascent proteins at different stages of folding. How the earliest steps of cotranslational folding (1$^{st}$ helix formation) could influence the ultimate nascent-chain dynamics remains a question for future investigations.

The 4xA mutations destabilize the native hydrophobic packing of the domain and have a significant effect on native tertiary interactions. The FPA results show that nascent wt and 4xA NTDs form very similar compact intermediates inside the exit tunnel. PET-FCS analysis indicates that the dynamic fluctuations between compact and dynamic states have identical transition-state barriers ($\Delta G^{\ddagger}$=37.5 kJ mol$^{-1}$) for wt and 4xA HemK70 (*Figure 5c*; *Supplementary file 1* - table 8) and, in both cases, the respective intermediates are highly dynamic. In contrast, outside the tunnel, mutations in the hydrophobic core or loop extensions increase dynamic fluctuations of the nascent chains (*Figure 5c,d*; *Supplementary file 1* - table 8). In particular for the stably folded HemK112, the effect of the mutations is very large, decreasing the $\Delta G^{\ddagger}$ value by ~10 kJ mol$^{-1}$ (*Figure 5c,d*). The free-energy landscapes of protein folding in solution are generally shallow, i.e. the differences between the highs (barriers) and lows (energy minima) are in the tens of kJ mol$^{-1}$ rather than hundreds (*Gruebele et al., 2016*). For example, the differences between partially unfolded high-energy states and the native states of proteins are between 17–54 kJ mol$^{-1}$ (*Englander and Mayne, 2014*). In the case of HemK NTD, a 10 kJ mol$^{-1}$ increase in the folding transition-state barrier is sufficient to slow the nascent-chain fluctuations by at least two orders of magnitude (*Figure 4e,f*; *Supplementary file 1* - table 5).

Such Eyring transition-state barrier analysis does not account for the diffusive nature of protein folding that is subject to solvent friction, internal chain friction and multiple stochastic attempts at crossing the transition-state energy barrier (*Zhou, 2010*; *Hagen, 2010*). Certain experimental designs allow for the application of a more encompassing Kramers rate theory for diffusion-controlled reactions to more rigorously characterize protein folding (*Nettels et al., 2007*; *Hagen, 2010*; *Schuler and Eaton, 2008*). However, to meaningfully apply the Kramers rate theory to our data we would have to include information about the solvent viscosity conditions (*Hagen, 2010*) which is not feasible given that we study protein folding inside the ribosome exit tunnel where conditions are thought to be unlike those of bulk solvent and probably change with peptide chain length (*Liutkute et al., 2020*). Understanding how the ribosome affects the multidimensional folding energy surface of the growing nascent peptide remains one of the more challenging questions in the field of cotranslational protein folding.

In summary, the present work shows how a small α-helical protein domain folds cotranslationally. It starts folding as soon as the first helical elements pass the constriction of the exit tunnel of the ribosome. With growing nascent chain, the emerging helical segments dock onto each other sequentially. The folding pathway entails numerous intermediates that continue to rearrange even when the domain emerges from the ribosome. Inside the exit tunnel or in the proximity of the ribosome the nascent peptide is highly dynamic, undergoing structural fluctuations on the μs time scale. The fluctuations slow down as the domain moves away (or is released) from the ribosome. Destabilizing mutations have little effect on folding within the exit tunnel, but abolish the domain stabilization after its separation from the ribosome. The results show the power of FPA and PET-FCS in solving the trajectory of cotranslational protein folding and in characterizing the dynamic properties of folding intermediates.

# Materials and methods

**Key resources table**

| Reagent type (species) or resource | Designation | Source or reference | Identifiers | Additional information |
|---|---|---|---|---|
| Recombinant DNA reagent | HemK wt plasmid | DOI: 10.1126/science.aad0344 | ID_ECBD_2409 | |
| Recombinant DNA reagent | HemK 4xA plasmid | DOI: 10.1126/science.aad0344 | | |
| Recombinant DNA reagent | HemK looped plasmid | This paper | | Methods: HemK constructs for PET and FPA; *Supplementary file 1* - table 10 |
| Recombinant DNA reagent | HemK 4xA FPA plasmid | This paper | | Methods: HemK constructs for PET and FPA; *Supplementary file 1* - table 11 |
| Recombinant DNA reagent | HemK wt FPA plasmid | This paper | | Methods: HemK constructs for PET and FPA; *Supplementary file 1* - table 11 |
| other | Purified *E. coli* in vitro translation system | DOI: 10.1073/pnas.92.6.1945; 10.1021/ja407511q; 10.1126/science.1229017; 10.1126/science.aad0344; 10.1016/S0076-6879 (07)30001–3 | | |
| Chemical compound, drug | BODIPY FL NHS-Ester | Thermo Fisher Scientific | ID_D2184 | |
| Chemical compound, drug | ATTO 655-NHS-Ester | ATTO-Tec GmbH, Siegen, Germany | ID_AD 655–31 | |
| Software, algorithm | Prism GraphPad | GraphPad Software, La Jolla California USA, www.graphpad.com | | |
| Software, algorithm | KinTek Explorer, version 6.3 | KinTek Global Kinetic Explorer V 6.3 https://kintekcorp.com/software | | |
| Software, algorithm | MicroTime 200 system | MicroTime 200 system, PicoQuant, Berlin, Germany | | |
| Software, algorithm | SymPhoTime 64 version 2.2 | PicoQuant, Berlin, Germany | | |

## HemK constructs for PET and FPA

The HemK (methyl transferase HemK) coding sequence (ECBD_2409, 834 p, 277aa) was derived from the pET-24a vector. For PET measurements, Trp at position 78 and – where indicated – Trp6 were mutated to Phe either in the wt or the 4xA HemK NTD sequence. The looped construct was generated using Gibson assembly reaction protocol (*Gibson et al., 2010*; *Gibson et al., 2009*) introducing five additional glycines in the loop between helix 3 and helix 4, before the wt Gly at position 43. Primers used to generate the glycine insert: forward 5'-CTCGCCTTTGGCGGCGGC and reverse 5'-GCGTTTCACCGCCGCCGCC, primers used to linearize the vector: forward 5'-GGCGGCGGCGGCGGCGGTGAAACGCAGCT and reverse 5'-AAAGGCGAGGATAAAAGTACGCCCTTTGCC. mRNA transcription templates were generated for all lengths using universal commercially available T7 forward primer (Eurofins Genomics, Ebersberg) and three unique reverse

primers for the required constructs: 70 (5'-ATGAGCAATGGGTTCACCATCG), 102 (5'-TGCCTGC TCCACCAGACACTCC), 112 (5'-ACGGCAAGGTTGTTCAGGCA) (Eurofins Genomics, Ebersberg). All PET-FCS constructs are shown in *Supplementary file 1* - table 10.

The FPA reporters contained a sequence of wt or 4xA HemK (aa 1–101) or HemK Pro (1-52) (Q4P; R8P; R22P; L27P; G35P; L40P) followed by the 17 aa SecM stalling peptide (*Nakatogawa and Ito, 2002*), and a 20 aa sequence of of cold shock protein A (CspA) (aa 1–20) (UniProt ID: P0A9 × 9), all truncations shown in *Supplementary file 1* - table 11. The full-length plasmids were synthesized by Eurofins Genomics (Ebersberg, Germany) using a pEX-A128 vector carrying a kanamycin resistance cassette. C-terminal HemK truncations were performed via a cloning protocol that involved vector linearization during PCR (*Supplementary file 2*). All generated constructs were verified by Sanger sequencing (Microsynth AG, Göttingen, Germany). mRNA transcription templates were generated for all FPA constructs using universal T7 forward and CspA reverse primer (5'-AGGAGTGA TGAAGCCGAAGCCT) (Eurofins Genomics, Ebersberg).

All mRNAs lacked a stop codon and were transcribed in vitro in buffer (40 mM Tris-HCl, pH 7.5, 15 mM MgCl$_2$, 2 mM spermidine, 10 mM NaCl, 10 mM DTT, 5 mM GMP). The DNA template (10% (v/v)) was incubated with 3 mM each of GTP, ATP, CTP and UTP, pyrophosphatase (5 u/ml), Ribo-Lock RNase inhibitor (1.5% (v/v), Fermentas), and T7 RNA-polymerase (1.6 u/μL), for 4 hr at 37°C. The mRNA was purified by anion exchange chromatography on a HiTrap Q HP column (GE Health-care) operated on Äkta FPLC system in buffer 30 mM Bis-Tris pH 6, 1 mM EDTA, 300 mM NaCl. mRNA was eluted using a linear gradient from 300 mM to 1.5 M NaCl over 20 column volumes. The mRNA-containing fractions were pooled, the mRNAs precipitated with ethanol and the mRNAs pelleted by centrifugation at 4000 g for 1 hr at 4°C. The mRNA pellets were resuspended in RNase- and DNase-free water and the concentration was measured using Nanodrop 2000c (Thermo Scientific). 8 M urea 10% polyacrylamide gel electrophoresis was used for mRNA quality control.

## In vitro translation

Translation components, including 70S ribosomes, initiation factors, elongation factors (EF-G and EF-Tu) and total aminoacyl-tRNAs (aa-tRNAs) were prepared as described (*Holtkamp et al., 2015*; *Rodnina and Wintermeyer, 1995*; *Mittelstaet et al., 2013*; *Doerfel et al., 2013*; *Milon et al., 2007*). Initiation complexes were formed in buffer A (50 mM HEPES pH 7.5, 70 mM NH$_4$Cl, 30 mM KCl, 7 mM MgCl$_2$, with 2 mM DTT, and 2 mM GTP). Ribosomes (0.5 μM) were incubated with initiation factors (IF1, IF2, and IF3; 2.25 μM each), mRNA (1.5 μM), and ATTO655-[$^3$H]Met-tRNA$^{fMet}$ (1 μM; for PET-FCS) or BodipyFL-[$^3$H]Met-tRNA$^{fMet}$ (1 μM; for arrest peptide assay) for 45 min at 37°C. Fluorescence-labeled tRNAs were prepared as described (*Mittelstaet et al., 2013*). EF-Tu–GTP was prepared in buffer A by incubating EF-Tu–GDP (120 μM) with phosphoenol pyruvate (3 mM) pyruvate kinase (0.05 mg/mL) for 15 min at 37°C. The ternary complex EF-Tu–GTP–aa-tRNA was formed by adding total aminoacyl-tRNA (200 μM) to EF-Tu–GTP followed by a 1 min incubation at 37°C.

All in vitro translation reactions were performed in buffer B (50 mM HEPES pH 7.5, 70 mM NH$_4$Cl, 30 mM KCl, 3.5 mM MgCl$_2$, 1 mM DTT, 0.5 mM spermidine and 8 mM putrescine). Initiation complexes (40 nM) were mixed with EF-Tu–GTP–aa-tRNA (50 μM) and EF-G (1 μM), and incubated for 5 min at 37°C. All mRNAs lacked the final stop codon and produced RNCs stalled with a pep-tidyl-tRNA in the P site. For PET-FCS, the RNCs were purified from translation factors and unbound fluorescence-labeled tRNA by sucrose cushion centrifugation using 2.2 M sucrose in buffer B. The ribosomes were pelleted using the TLA-100 rotor (Beckman Coulter) at 68,000 rpm for 40 min at 4°C. The pellet containing RNC was resuspended in buffer B, and the RNC concentration was deter-mined by liquid-liquid radioactivity counting of $^3$H-labeled Met. The complexes were flash frozen in liquid nitrogen and stored in - 80°C until use. Translation efficiency was monitored on Tris-tricine SDS PAGE.

All translation samples were prepared for Tris-tricine SDS PAGE as follows. The nascent chains were released from the ribosome by adding 1.5 M hydroxylamine and incubating the samples for 1 hr at 37°C. The samples were then diluted with gel loading buffer (50 mM Tris-HCl pH 6.8, 4% w/v SDS, 2% v/v 2-mercaptoethanol and 12% w/v glycerol) and translation products were separated using Tris-Tricine SDS-PAGE (*Schagger, 2006*). A 16.5% separating gel (49.5% T, 6% C), 10% spacer gel and 4% stacking gel were used. The in vitro translation products were visualized by detecting the N-terminal dye using a Fujifilm FLA-9000 fluorescence scanner equipped with a laser of 488 nm wavelength to detect Bodipy FL, or the 680 nm laser to detect ATTO-655. The band intensities on

the gel were quantified and analyzed using LI-COR Biosciences GmbH Image Studio version 5.2.5. In case of 4xA variant FPA construct 65 onwards, the FL and AR products run as double bands and were quantified as such. The fraction of full-length product was calculated by dividing the full-length band intensity by the sum of the full-length and arrested band intensities (*Figure 1—figure supplement 1e*).

To produce free peptide, nascent chains were released from the ribosome with hydroxylamine. RNC was incubated with hydroxylamine (5% w/v; pH ≤8) for 1 hr at 37°C. This was followed by a sucrose cushion centrifugation as above. The supernatant containing the released nascent chains was collected for PET-FCS measurements.

## Fluorescence correlation spectroscopy

Fluorescence correlation measurements were performed using the MicroTime 200 system (Pico-Quant, Berlin, Germany), which is based on a modified Olympus IX 73 confocal microscope and equipped with a water objective lens with 60x magnification and 1.2 N.A. (Olympus UPlanSApo). For excitation, a collimated laser (LDH-D-C-640, PicoQuant GmbH) beam with 636.5 nm wavelength (operated in continuous wave mode) with large diameter was focused through the objective into the sample solution. The laser power was set to ~40 µW to minimize Atto655 triplet state formation and to avoid photobleaching. Fluorescence signals were collected using the same objective (epifluorescence configuration) and separated from the excitation light by a dichroic mirror. After that, the collected fluorescence light was focused through a 50 µm pinhole to eliminate fluorescence coming from axial positions away from the focal plane (confocal detection). A 50/50 beam splitter was used to split the fluorescence signal into two channels, where light was focused onto two single-photon avalanche photodiodes (SPAD) after passing through a band pass filter (690/70 nm). The signals of two SPADs were cross-correlated to eliminate SPAD after-pulsing effects.

Purified RNCs were measured at ~4 nM in buffer B, and sample concentration was adjusted in such a way as to yield an average of one molecule within the confocal detection volume for all measurements. For the Trp titrations, ACFs were recorded for purified RNCs of HemK 70 W6F and HemK 102 W6F at different concentrations of added Trp. A solution of 70 mM Trp was prepared in buffer B, and final free Trp concentrations ranged from 1.8 mM to 45 mM. The recorded ACFs were then fitted (*Equation 1*), and the parameters obtained were processed as described below. Measurements were performed at ambient temperature (22°C). For each RNC solution, single-photon fluorescence detection events were recorded for at least four consecutive time intervals of 10 min. The auto-correlation functions (ACF) were computed using the SymPhoTime 64 software (PicoQuant). After normalization, these ACFs were compared to confirm that the RNCs were stable throughout the duration of the measurement, and the technical replicates were averaged. The experiments were repeated a minimum of 2–3 times for the same class of RNCs from different preparations.

The microscope is set up with a detection volume where the longitudinal dimension is much larger than the transverse dimension. The variance in fit using a 3D versus 2D diffusion models for the fast relaxation times was 2–9%, and 11% for $\tau_D$, therefore fitting of initial ACFs was carried out using a model for 2D single species diffusion (*Krichevsky and Bonnet, 2002*; *Allen and Thompson, 2006*) with two relaxation rate constants, a triplet rate constant, and a diffusion rate constant,

$$G(\tau) = \left(1 + c_1 e^{-k_1 t} + c_2 e^{-k_2 t}\right)\left(\frac{1 - F + F\, e^{-k_f t}}{1 - F}\right)\left(\frac{1}{N}\right)(1 + k_d t)^{-1}, \tag{1}$$

where $k_1$ and $k_2$ are apparent relaxation rate constants with respective amplitudes $c_{1\ and\ 2}$, $N$ is the average number of molecules in the confocal volume, $F$ is the amplitude for the triplet component with rate constant $k_f$, and $k_d$ is the inverse diffusion time.

## Kinetic modeling

To fit the ACFs of the PET-FCS measurements in the commercial KinTek software, the signal from diffusion and triplet state components was removed from each curve using the respective fitted parameters (*Equation 1*; *Supplementary file 1* - table 2). The software KinTek Global Kinetic Explorer V 6.3 was used for kinetic modeling (*Johnson et al., 2009a*; *Johnson et al., 2009b*). In short, the KinTek software is specifically designed for kinetic analysis of complex biological reactions and upon defining the model in simple alphanumerical terms (states); the software automatically

generates the differential equations and the numerical integration without simplification or further user input (*Johnson et al., 2009a*). Using this software, we are able to test many different models for nascent-chain dynamics and by utilizing the multiple rigorous statistical analysis tools integrated into the software (*Johnson et al., 2009b*) we can immediately analyze the quality of the results in a fast and robust way. In all cases, the exponential decays of different PET quenching curves were simulated by the KinTek software as a sum of species C, D, R (or Rx), and W (or Wx) (see text) for a given HemK construct, multiplied by a species-specific amplitude coefficient. The amplitude coefficients were assumed to be identical for all simulated traces in a particular dataset. We determined the model that was sufficient to fit our data by testing a series of models with increasing complexity:

Model 2e is two-equilibria model (*Figure 4—figure supplement 1a*) that describes three nascent-chain states: the nascent-chain interactions with the ribosome (state D ↔ R), and the nascent- chain interaction with Trp6 (D ↔ W). This model could not fit the data in a satisfactory way as indicated by the poor quality-of-fit, reflected by a $\chi^2$/DoF value of 39 (*Figure 4—figure supplement 1c*).

Model 3e is three-equilibria model that introduces one additional reversible reaction to model 2e to describe conformational fluctuations between states (D ↔ C) (*Figure 4—figure supplement 1b*). This model was also insufficient to obtain a good quality fit, reflected by a $\chi^2$/DoF value of 46 (*Figure 4—figure supplement 1d*).

Model 5e-O - contains five-equilibria: conformational fluctuations between different states of nascent chains (C ↔ D), in each of these states the nascent chain can be quenched by the ribosome surface (R) or by the in-chain Trp (W), meaning that there are two additional equilibria C ↔ R and C ↔ W. Here the quenched state R and state W connect the two states C and D and allow the conversion between nascent-chain conformational states via one of the quenched states (*Figure 4—figure supplement 2*).

Model 5e-H – includes the same equilibria as model 5e-O, but separates the ribosome- and Trp-quenched states to Rc, Rd and Wc, Wd, respectively, and postulates that the conformational fluctuations can only occur directly between states (*Figure 4*). Both 5e models were utilized to successfully fit the HemK RNC ACF data, reflected by a $\chi^2$/DoF value of 1.5 for model 5e-O, and 3.7 for model 5e-H (*Figure 4*, *Figure 4—figure supplements 2* and *4*).

The dominant Trp quenching mechanism for ATTO655 is through static quenching due to stacking interactions in the fluorophore–quencher pair; the dissociation rate of the quenched complex is defined by the specific properties of these stacking interactions (*Limpouchova and Prochazka, 2016*; *Sharma et al., 2017*). Therefore, the dissociation rate (dequenching) constant of the ATTO655–Trp complex is expected to be independent of the source of Trp or the structure of the RNC. To estimate the $k_{off}$ for the ATTO655–Trp complex, we measured ACFs for HemK70 W6F and HemK102 W6F RNCs with increasing concentrations of free Trp in solution (*Figure 4—figure supplement 3*). Each RNC construct contained 7 ACFs for each tryptophan concentration of [Trp]=0, 1.8, 4.5, 9, 18, 27, 45 mM (dataset A), and each of these data curves contained the average of two independent replicates. The amplitudes of reactions in the ns and μs time domain increased with free Trp concentrations.

To determine the dequenching constant, we globally fit the titration dataset A to the kinetic models 5e-H and 5e-O (*Figure 4—figure supplement 3*) after introducing a term that is concentration-dependent (Trp binding to ATTO655), while all other rate constants remained concentration-independent and constant in all of these experiments. Global fitting of the data to model 5e-H yielded a $2.2 \pm 0.2\ \mu s^{-1}$ dequenching rate constant of the ATTO655–Trp pair (*Supplementary file 1* - table 3), while model 5e-O yielded a $2.0 \pm 0.1\ \mu s^{-1}$ dequenching rate (*Supplementary file 1* - table 4). Other rates were not sufficiently constrained by the Trp titration dataset, as evident from large standard errors (*Supplementary file 1* - tables 3 and 4), and were not used in the following fitting. As the dissociation rate is defined by the specific properties of these stacking interactions, the dissociation rate constant of the ATTO655–Trp complex is expected to be independent of the identity of Trp or the structure of the RNC, which allowed us to lock the respective rates for fitting of the dataset for different RNCs.

In the second step, we globally fit the experimental dataset for different RNCs: 70, 102, 112 aa-long HemK wt with Trp (W6) and without Trp (W6F), the 70 and 112 HemK 4xA W6 and W6F, HemK 102 loop W6F and HemK 112 loop W6F curves (dataset B) to the two 5e models. To reduce the number of independent fitting parameters, we locked the ATTO655–Trp dissociation rates at 2 $\mu s^{-1}$ and at 2.2 $\mu s^{-1}$ in their respective model 5e-O and 5e-H (W and Wx state $k_{off}$ values), as described

above. Also the $k_{off}$ rate from state R or Rx were linked across all constructs, as this rate also depends on the properties of the dye-quenching pair, which should be uniform across the different complexes (light green in *Supplementary file 1* - tables 5 and 6). We linked the $k_{on}$ values pairwise for the wt and 4xA constructs for HemK 70, for example, for the D $\leftrightarrow$ R/Rd transition the $k_{on}$ values for the wt and 4xA were linked, because they likely depend on the position of the nascent chain in the tunnel and were very similar in our initial fitting. For all constructs (except 112 4xA), transitions from state C to R/Rx and state C to W/Wx $k_{on}$ were also linked, because initial fitting suggests that they are small and very similar. The loop construct R/Rx state $k_{on}$ rates were linked to the corresponding rates for the wt construct R/Rx, again because these rates are likely to be similar for the RNCs of similar length. In the case of HemK112 4xA, all rates were separated from other constructs except the $k_{off}$ of R/Rx states. All reported $\chi^2$/DoF values and standard errors of mean in *Supplementary file 1* - tables 3, 4, 5 and 6, were calculated using a covariance matrix derived using nonlinear regression algorithms.

Complete fitting results to model 5e-H are displayed in *Figure 4*, *Figure 4—figure supplements 3* and *4*; *Supplementary file 1* - tables 3, 5 and 8. Complete fitting results to model 5e-O are displayed in *Figure 4—figure supplements 2*, *3* and *4*, *Supplementary file 1* - tables 4, 6 and 9. The confidence intervals for rate values obtained by the kinetic fits were calculated by the KinTek software using their in-built fitspace quality-of-fit assessment algorithm (*Johnson et al., 2009b*) that examines confidence contours at a constant $\chi^2$ value of 0.833 for each rate parameter (*Supplementary file 1* - table 7).

## Calculations of the transition-state energy barrier

Transition-state theory was used to calculate the energy barrier between the conformational state ensembles of different HemK constructs. The $k_{on}$ and rates of conformational fluctuations between states that were obtained from the kinetic modeling were used to solve for the transition state ($\Delta G^{\ddagger}$) energy barrier between the states (*Fersht et al., 1999*; *Baryshnikova et al., 2005*)

$$k_{on} = k_B \frac{T}{h} \kappa \cdot \exp\left(\frac{-\Delta G_{\ddagger-D}}{RT}\right), \tag{2}$$

where *R* is the gas constant 8.3145 J mol$^{-1}$; *T* is temperature (295˚K); $\kappa$ is transmission coefficient (approximated to 1.0 in transition-state theory; *Fersht et al., 1999*); $k_B$ is Boltzmann's constant 1.38 $\times 10^{-23}$ J K$^{-1}$; and *h* is Planck's constant 6.6 $\times 10^{-34}$ m$^2$ kg s$^{-1}$.

## Acknowledgements

We thank Wolfgang Wintermeyer for critical reading the manuscript; Anna Pfeifer, Olaf Geintzer, Sandra Kappler, Christina Kothe, Theresia Niese, Tanja Wiles, Franziska Hummel, Tessa Hübner, Vanessa Herold, and Michael Zimmermann for expert technical assistance. Funding: The work was funded by the Max Planck Society and by the European Research Council (ERC) Advanced Investigator Grant RIBOFOLD (proposal number n˚ 787926).

## Additional information

### Funding

| Funder | Grant reference number | Author |
| --- | --- | --- |
| European Research Council | RIBOFOLD Nr. 787926 | Marina V Rodnina |
| Max-Planck-Gesellschaft | | Marina V Rodnina |

The funders had no role in study design, data collection and interpretation, or the decision to submit the work for publication.

### Author contributions

Marija Liutkute, Conceptualization, Data curation, Formal analysis, Validation, Investigation, Visualization, Methodology, Writing - original draft, Writing - review and editing; Manisankar Maiti, Data

curation, Formal analysis, Validation, Investigation, Methodology; Ekaterina Samatova, Conceptualization, Supervision, Methodology, Writing - review and editing; Jörg Enderlein, Software, Supervision, Methodology, Writing - review and editing; Marina V Rodnina, Conceptualization, Resources, Supervision, Funding acquisition, Methodology, Project administration, Writing - review and editing

Author ORCIDs
Marija Liutkute  https://orcid.org/0000-0002-3462-7472
Jörg Enderlein  http://orcid.org/0000-0001-5091-7157
Marina V Rodnina  https://orcid.org/0000-0003-0105-3879

Decision letter and Author response
Decision letter https://doi.org/10.7554/eLife.60895.sa1
Author response https://doi.org/10.7554/eLife.60895.sa2

## Additional files

### Supplementary files
• Supplementary file 1. Supplementary files of fitting result numerical values and amino acid sequences of analyzed HemK constructs. Table 1. ACF (each average of $N \geq 8$) fits of HemK constructs in solution. Table 2 results of empirical fits of PET-FCS ACF (each ACF an average of $N \geq 8$) for RNCs. Table 3 results of global fitting of the free Trp titration (dataset A) to model 5e-H. Table 4 results of global fitting of the free Trp titration (dataset A) to model 5e-O. Table 5. Results of global fitting of the dataset B to the model 5e-H. Table 6. Results of global fitting of the dataset B to the model 5e-O. Table 7. Upper and lower boundaries of each rate parameter of model 5e-H and 5e-O global fitting. Table 8. Free-energy calculations for all RNC constructs using rates derived from the 5e-H model. Table 9. Free-energy calculations for all RNC constructs from the 5e-O model rates. Table 10. PET-FCS constructs aa sequences N- to C-terminus. Table 11. Force-profile construct of wt HemK full-length, aa sequence N- to C-terminus.

• Supplementary file 2. Primer list for the force-profile constructs of HemK wt/4xA.

• Transparent reporting form

### Data availability
Data analysed in this study are displayed in figures of the main text and source data files are provided for figures 1,2 and 3.

The following previously published datasets were used:

| Author(s) | Year | Dataset title | Dataset URL | Database and Identifier |
|---|---|---|---|---|
| Nilsson OB, Hedman R, Marino J, Wickles S, Bischoff L, Johansson M, Muller-Lucks A, Trovato F, Puglisi JD, O'Brien EP, Beckmann R, von Heijne G | 2015 | Single-particle cryo-EM of co-translational folded adr1 domain inside the *E. coli* ribosome exit tunnel | https://www.ebi.ac.uk/pdbe/entry/emdb/EMD-3079 | Electron Microscopy Data Bank, EMD-3079 |
| Yang Z, Shipman L, Zhang M, Anton BP, Roberts RJ, Cheng X | 2004 | Crystal Structure Analysis of *E.coli* Protein (N5)-Glutamine Methyltransferase (HemK) | https://www.rcsb.org/structure/1T43 | RCSB Protein Data Bank, 1T43 |

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
