## [Decision Letter]

**Acceptance summary:**

Liutkute et al. investigate the co-translational folding of a small α-helical domain from HemK using biochemical assays of "force" on the nascent protein and spectroscopic assays of intramolecular dynamics with an N-terminal fluorescent probe. The results of these experiments provide a high-resolution view of the sequential folding path of a small model protein within and outside the ribosome. The study provides an important foundation for understanding the molecular dynamics of co-translational protein folding.

**Decision letter after peer review:**

Thank you for submitting your article "Gradual Compaction of the Nascent Peptide During Cotranslational Folding on the Ribosome" for consideration by *eLife*. Your article has been reviewed by three peer reviewers, one of whom is a member of our Board of Reviewing Editors, and the evaluation has been overseen by Cynthia Wolberger as the Senior Editor. The reviewers have opted to remain anonymous.

The reviewers have discussed the reviews with one another and the Reviewing Editor has drafted this decision to help you prepare a revised submission.

Summary:

This study by Liutkute et al. investigates the co-translational folding of a small α-helical domain from HemK. The study continues earlier work by Rodnina and colleagues that showed using FRET and other measurements that HemK begins folding inside the ribosome exit tunnel and occurs sequentially as individual α-helical segments are able to be accommodated in the exit tunnel vestibule. Folding completes just outside the ribosome when the entire HemK domain is exposed. The current work extends these earlier studies using biochemical assays of "force" on the nascent chain and spectroscopic assays of intramolecular dynamics with an N-terminal fluorescent probe. The force assays illustrate that tension is seen as individual α helices move beyond the exit tunnel constriction, and at other previously documented steps of folding in the vestibule. These intra-ribosomal events are not impacted by a mutation that disrupts packing of the hydrophobic core. The fluorescence quenching dynamics using PET-FCS on HemK nascent chains reveal two interconverting states, compact (C) and dynamic (D). Both states are populated similarly regardless of chain length. However, the barrier between these states increases when the domain emerges from the ribosome. These experiments indicate a destabilizing effect of the ribosome on the nascent chain, presumably due to spatial constraints that are imposed. Taken together, the experiments support earlier work that proposed a sequential folding path in which the order of helix formation of the 5-helix NTD of HemK follows the order from N- to C-terminus.

Essential revisions:

The reviewers agreed that the experiments were carefully performed and interpreted. The primary experimental revision relates to providing a suitable negative control for the FPA assay and being explicit about its interpretation. Suitable characterization of this assay is important because, although the assay itself is not new, this appears to be the first time it is being used to analyze folding at single codon resolution.

1) The changes observed in the force profile that are ascribed to folding events are only meaningful if the authors can fully document that an unstructured segment of polypeptide showed a relatively flat and low-tension force profile when analyzed at comparable single-codon resolution. Such an experiment would illuminate the degree of folding-independent variability caused by other events such as non-specific nascent chain interactions with the exit tunnel wall, dynamic fluctuations, and so forth.

2) A complement to the negative control requested in point (1) would be a demonstration that another well-characterized folded domain generates "force" in this assay. The authors could, for example, test some of the spectrin constructs from Kemp et al., 2020, or some of the constructs from Farías-Rico et al., 2018. The reviewers realize that extensive analysis of another substrate is beyond the scope of this study, so this is offered as an optional suggestion to more convincingly correlate folding events to relief of stalling.

3) The authors should be explicit about what is being measured in the "force" sensor assay. SecM stalling relies on a specific secondary structure of the stalling sequence that causes an altered P site geometry that is unfavourable for peptide bond formation. Stalling will not occur if this altered geometry cannot be stabilized. Thus, what the authors refer to as 'force' is actually a constraint applied to the nascent chain to prevent SecM secondary structure formation. Thus, the folding is not generating force so much as constraining the nascent chain as a consequence of the ribosome exit tunnel geometry. It is a subtle, but I feel important, distinction to explain the assay. The reason is that such a constraint can be due to reasons other than folding. For example, an interaction between the nascent chain and the exit tunnel (or other proteins) could similarly constrain the nascent chain. Such alternative interpretations of the assay are important caveats to include in the text. Properly explaining the basis of this assay would help to rationalize how "force" can be generated by a single α helix.

---

## [Author Response]

Essential revisions:The reviewers agreed that the experiments were carefully performed and interpreted. The primary experimental revision relates to providing a suitable negative control for the FPA assay and being explicit about its interpretation. Suitable characterization of this assay is important because, although the assay itself is not new, this appears to be the first time it is being used to analyze folding at single codon resolution.1) The changes observed in the force profile that are ascribed to folding events are only meaningful if the authors can fully document that an unstructured segment of polypeptide showed a relatively flat and low-tension force profile when analyzed at comparable single-codon resolution. Such an experiment would illuminate the degree of folding-independent variability caused by other events such as non-specific nascent chain interactions with the exit tunnel wall, dynamic fluctuations, and so forth.

To answer this major question, we have performed the force profile assay (FPA) experiments on a variant of HemK that contains multiple proline substitutions in the N-terminal helixes (Q4P; R8P; R22P; L27P; G35P; L40P)(Figure 1—figure supplement 1F, G). Prolines are known to perturb α-helical structures and we expect that these mutations alter the native folding of HemK N-terminal domain. We generated the HemK constructs in two amino acid steps and followed the tension in the nascent chain when the domain is within the ribosome exit tunnel. We demonstrate that the tension profile in the nascent chain in regions I – III becomes relatively flat and low-tension (Figure 1—figure supplement 1F, G). This should answer the question concerning the contribution of folding to the high-tension regions caused by the events in the tunnel. The respective changes were introduced in the text in the Results and in the Discussion.

2) A complement to the negative control requested in point (1) would be a demonstration that another well-characterized folded domain generates "force" in this assay. The authors could, for example, test some of the spectrin constructs from Kemp et al., 2020, or some of the constructs from Farías-Rico et al., 2018. The reviewers realize that extensive analysis of another substrate is beyond the scope of this study, so this is offered as an optional suggestion to more convincingly correlate folding events to relief of stalling.

The comparison between FPA profiles of HemK and domain of a different fold, FLN5, has been reported by von Heijne’s group ^1^. Their work demonstrated that the results of the SecM arrest peptide-mediated FPA correlates well with fluorescent and biophysical methods used to analyze folding events on the ribosome. The HemK results we obtain here are comparable with constructs used by von Heijne and colleagues ^1^. Given the general agreement between our and von Heijne’s results, we feel that the question raised by the reviewers has been satisfactorily answered already ^1-5^. This is now more clearly stated in the Discussion.

3) The authors should be explicit about what is being measured in the "force" sensor assay. SecM stalling relies on a specific secondary structure of the stalling sequence that causes an altered P site geometry that is unfavourable for peptide bond formation. Stalling will not occur if this altered geometry cannot be stabilized. Thus, what the authors refer to as 'force' is actually a constraint applied to the nascent chain to prevent SecM secondary structure formation. Thus, the folding is not generating force so much as constraining the nascent chain as a consequence of the ribosome exit tunnel geometry. It is a subtle, but I feel important, distinction to explain the assay. The reason is that such a constraint can be due to reasons other than folding. For example, an interaction between the nascent chain and the exit tunnel (or other proteins) could similarly constrain the nascent chain. Such alternative interpretations of the assay are important caveats to include in the text. Properly explaining the basis of this assay would help to rationalize how "force" can be generated by a single α helix.

SecM arrest peptide mediated force sensing method has been previously characterized in a number of publications ^1-4,6-10^, and the relationship between applied force (up to 60 pN) to the nascent chain and the fraction of relieved ribosomes from SecM stalled constructs has also been specifically investigated ^10^. By performing the experiments with HemK Pro variant (Figure 1—figure supplement 1F, G) we show that the high-tension regions that we see when nascent chain is inside the ribosome exit tunnel come from cotranslational folding events. The peaks II and III largely disappear when helix folding is altered. Some residual amplitude indicates the possibility that SecM stalling could be only marginally alleviated by other events, such as interactions between the nascent chain and the exit tunnel walls, random configurations of the nascent chain inside the exit tunnel, or nonspecific compaction events in the nascent chain. This is now more fully referred to in the Introduction, Results and Discussion.

References:

1) Kemp, G., Kudva, R., de la Rosa, A. and von Heijne, G. Force-profile analysis of the cotranslational folding of HemK and filamin domains: Comparison of biochemical and biophysical folding assays. J Mol Biol 431, 1308-1314 (2019).2) Farias-Rico, J.A., Ruud Selin, F., Myronidi, I., Fruhauf, M. and von Heijne, G. Effects of protein size, thermodynamic stability, and net charge on cotranslational folding on the ribosome. Proc Natl Acad Sci U S A 115, E9280-E9287 (2018).3) Nilsson, O.B. et al. Cotranslational Protein Folding inside the Ribosome Exit Tunnel. Cell Rep 12, 1533-40 (2015).4) Ismail, N., Hedman, R., Schiller, N. and von Heijne, G. A biphasic pulling force acts on transmembrane helices during translocon-mediated membrane integration. Nat Struct Mol Biol 19, 1018-22 (2012).5) Cabrita, L.D. et al. A structural ensemble of a ribosome-nascent chain complex during cotranslational protein folding. Nat Struct Mol Biol 23, 278-285 (2016).6) Tian, P. et al. Folding pathway of an Ig domain is conserved on and off the ribosome. Proc Natl Acad Sci U S A 115, E11284-E11293 (2018).7) Su, T. et al. The force-sensing peptide VemP employs extreme compaction and secondary structure formation to induce ribosomal stalling. Elife 6(2017).8) Nilsson, O.B. et al. Cotranslational folding of spectrin domains via partially structured states. Nat Struct Mol Biol (2017).9) Kemp, G., Nilsson, O.B., Tian, P., Best, R.B. and von Heijne, G. Cotranslational folding cooperativity of contiguous domains of alpha-spectrin. Proc Natl Acad Sci U S A 117, 14119-14126 (2020).10) Goldman, D.H. et al. Mechanical force releases nascent chain-mediated ribosome arrest in vitro and in vivo. Science 348, 457-460 (2015).